# Phthalate Exposure and Oxidative/Nitrosative Stress in Childhood Asthma: A Nested Case-Control Study with Propensity Score Matching

**DOI:** 10.3390/biomedicines10061438

**Published:** 2022-06-17

**Authors:** Jung-Wei Chang, Hsin-Chang Chen, Heng-Zhao Hu, Wan-Ting Chang, Po-Chin Huang, I-Jen Wang

**Affiliations:** 1Institute of Environmental and Occupational Health Sciences, School of Medicine, National Yang Ming Chiao Tung University, Taipei 112304, Taiwan; jungwei723@ym.edu.tw (J.-W.C.); eeyy.hu@gmail.com (H.-Z.H.); 2Department of Chemistry, Tunghai University, Taichung 407224, Taiwan; hsinchang@thu.edu.tw; 3National Institute of Environmental Health Sciences, National Health Research Institutes, Miaoli 35042, Taiwan; wtchang2@nhri.edu.tw; 4Department of Medical Research, China Medical University Hospital, China Medical University, Taichung 406040, Taiwan; 5Department of Safety, Health and Environmental Engineering, National United University, Miaoli 36003, Taiwan; 6Research Center for Environmental Medicine, Kaohsiung Medical University, Kaohsiung 80708, Taiwan; 7Department of Pediatrics, Taipei Hospital, Ministry of Health and Welfare, Taipei 10341, Taiwan; 8College of Public Health, China Medical University, Taichung 406040, Taiwan

**Keywords:** phthalates, oxidative stress, asthma, propensity score matching

## Abstract

Whether low-dose phthalate exposure triggers asthma among children, and its underlying mechanisms, remain debatable. Here, we evaluated the individual and mixed effects of low-dose phthalate exposure on children with asthma and five (oxidative/nitrosative stress/lipid peroxidation) mechanistic biomarkers—8-hydroxy-2′-deoxyguanosine (8-OHdG), 8-nitroguanine (8-NO_2_Gua), 4-hydroxy-2-nonenal-mercapturic acid (HNE-MA), 8-isoprostaglandin F2α (8-isoPF2α), and malondialdehyde (MDA)—using a propensity score-matched case-control study (case vs. control = 41 vs. 111). The median monobenzyl phthalate (MBzP) concentrations in the case group were significantly higher than those in the control group (3.94 vs. 2.52 ng/mL, *p* = 0.02), indicating that dust could be an important source. After adjustment for confounders, the associations of high monomethyl phthalate (MMP) (75th percentile) with 8-NO_2_Gua (adjusted odds ratio (aOR): 2.66, 95% confidence interval (CI): 1.03–6.92) and 8-isoPF2α (aOR: 4.04, 95% CI: 1.51–10.8) and the associations of mono-iso-butyl phthalate (MiBP) with 8-isoPF2α (aOR: 2.96, 95% CI: 1.13–7.79) were observed. Weighted quantile sum regression revealed that MBzP contributed more than half of the association (56.8%), followed by MiBP (26.6%) and mono-iso-nonyl phthalate (MiNP) (8.77%). Our findings supported the adjuvant effect of phthalates in enhancing the immune system response.

## 1. Introduction

Asthma is a leading chronic illness in children, imposing an increasing burden on healthcare systems. In Taiwan, its prevalence in the general population under 12 years old is 5.6% and above 12 years old is 22.6% [1]. Childhood asthma is associated with considerable school absence, family stress, and direct medical costs. Clear evidence is lacking regarding whether the cause of asthma is related to an individual’s genetic factors or whether it is only a manifestation of chronic exposure to confounding risk factors within environmental exposures (allergens and pet exposure) and demographics (obesity). Furthermore, some environmental factors, such as secondhand tobacco smoke exposure, result in higher asthma severity [2]. Maternal smoking can aggravate asthma symptoms and can cause asthma attacks in early childhood [3,4]. Family history of asthma and raising furry pets may be associated with a history of asthma in the same families, thereby indicating that they can be used as risk factors to identify patients with asthma [5]; nevertheless, a consistent finding between asthma and phthalate exposure is lacking. Hsu et al. (2012) reported that increasing monobenzyl phthalate (MBzP) exposure would increase asthma incidence, whereas levels of mono (2-ethylhexyl) phthalate (MEHP), a metabolite of di (2-ethylhexyl) phthalate (DEHP), were associated with the severity of allergic rhinitis [6]. By contrast, Callesen et al. observed a significant association only between DEHP and wheezing and not between household phthalates and doctor-diagnosed childhood asthma, rhinoconjunctivitis, or atopic dermatitis [7]. Additionally, no urinary phthalate metabolite has been found to be associated with asthma or rhinoconjunctivitis [8]; instead of being a causative agent, most phthalates may act as adjuvants for the agents that potentially cause allergic sensitization [9]. Overall, these epidemiological studies have observed inconsistent relationships between phthalate exposure and the incidence of allergic diseases.

A well-recognized mechanism in the pathogenesis of asthma is described by prooxidant-antioxidant defense systems imbalance, with increased generation of oxidative and nitrosative stress-related mediators in asthmatics [10,11]. The most important prooxidants are reactive oxygen species (ROS) and reactive nitrogen species (RNS), which can also contribute to the progression of existing airway inflammation [12]. In addition, accumulation of oxidative damage causes the damage of DNA, proteins, and lipids with decline in physiological functions, and airway structural changes [13]. Direct measurement of ROS levels with high accuracy and precision is difficult by their low concentrations, short lifespan, and rapid reactivity with the redox state; therefore, various indirect markers of oxidative damage, such as markers of lipid peroxidation (e.g., malondialdehyde (MDA), 8-isoprostane, acrolein, hexanal, 4-hydroxyhexanal, 4-hydroxy-2-nonenal-mercapturic acid (HNE-MA)), or DNA damage (e.g., 8-hydroxy-2′-deoxyguanosine (8-OHdG)), have be used [14].

Some in vivo and in vitro studies have revealed that phthalate exposure mediates the association among the enhancement of respiratory sensitization, allergic responses, and oxidative stress [15,16,17,18,19,20]. Dibutyl phthalate (DBP) exposure substantially increased the levels of oxidative stress biomarkers, namely, 8-hydroxy-2-deoxyguanosine (8-OHdG) and malonaldehyde (MDA), in an animal study [19]; this finding implies that these indicators can be used for evaluating the potential mediating role of oxidative stress in phthalate exposure-induced respiratory and allergic symptoms in patients with asthma [21].

In our previous study, urine phthalate metabolites were associated with oxidative stress-related genetic variants, such as superoxide dismutase 2 (SOD2). However, we could not test the hypothesis that oxidative stress biomarkers might modify the association of phthalate exposure with asthma directly [18]. Moreover, we only measured four urine phthalate metabolites: monoethyl phthalate (MEP), monobutyl phthalate (MBP), MBzP, and mono(2-ethyl-5-hydroxyhexyl) phthalate (MEHHP). Given that various mechanisms can induce respiratory and allergic diseases, it is necessary to explore the various oxidative stress biomarkers, such as 8-OHdG for DNA damage, as well as 8-nitroguanine (8-NO_2_Gua). Lipid peroxidation is also an important indicator of allergic responses, and several major products, such as urinary HNE-MA, 8-isoprostaglandin F2α (8-isoPF2α), and MDA, have often been evaluated simultaneously [22,23]. However, the association among phthalate exposure, risk of an asthma attack, and oxidative/nitrosative stress biomarkers is warranted to elucidate. Furthermore, whether the contribution from individual and simultaneous phthalate exposure to the increased incidence of asthma was different also needs to be evaluated.

To extend the aforementioned findings and verify the hypothesis, we evaluated (i) the potential individual and mixed effects of 11 phthalate metabolites using a logistic regression model in children with asthma; and (ii) the mediating effect of oxidative stress biomarkers—8-OHdG, 8-NO_2_Gua, HNE-MA, 8-isoPF2α, and MDA—on the association between phthalate exposure and asthma.

## 2. Materials and Methods

### 2.1. Study Population

This matched case–control study included children with asthma and controls. The anthropometric data of the children were measured and recorded. In addition, a standardized questionnaire was used to collect the basic demographics, birth history, family income, parental history of allergic diseases, and the child’s environmental exposure to pets and tobacco smoke, and responses were provided by the parents. After the sampling protocol was explained to the child and parents, written informed consent was obtained. The study protocol was approved by Taipei Hospital’s Institutional Review Board (TH-IRB-0020-0009).

### 2.2. Case Definition and Assessment of Confounders

To evaluate postnatal exposures such as furry pets, carpets, and environmental tobacco smoke exposure, the International Study of Asthma and Allergies in Childhood questionnaire was administered by an experienced pediatric allergist. The corresponding interview process was conducted according to the protocol in our previous study [24]. Moreover, doctor-diagnosed asthma was confirmed by pediatric allergists based on the following three criteria: (i) recurrence of at least two of three symptoms: cough, wheeze, and shortness of breath within the previous 12 months in absence of a cold; (ii) doctor’s diagnosis of asthma with ongoing treatment; and (iii) responding to therapy with β_2_-agonists or inhaled corticosteroids [25].

### 2.3. A Nested Case-Control Study with Propensity Score Matching

All children were selected from the Childhood Environment and Allergic Diseases Study (CEAS) cohort recruited in 2010 in Taipei [18,24]. Because asthma can be affected by some demographics (such as sex and BMI), we used propensity score matching to ensure a balanced covariate distribution between the patient and healthy control groups; the balance was examined using the standardized mean difference (SMD) and the variance ratio (VR). Thus, proper or balanced matching is evidenced as the SMD < 0.1 and the VR was 0.5–2. Three controls were matched with one case in case of the availability of urine specimens and complete demographic characteristic data.

### 2.4. Analysis of Oxidative/Nitrosative Stress Biomarkers

Simultaneous measurement of four specific oxidative/nitrosative stress biomarkers—8-OHdG, 8-NO_2_Gua, 8-isoPF2α, and HNE-MA—was performed using isotope dilution liquid chromatography-tandem mass spectrometry (LC-MS/MS), as described by Wu et al. [23], with some modifications.

The collected urine samples were thawed at 4 °C overnight after being stored at −20 °C. After centrifugation at 12,500× *g* for 15 min, 50 µL of the urine supernatant was diluted with 195 µL of Milli-Q water added with 1 mM ammonium acetate (AA) and then spiked with 5 µL of a mixture of isotope-labeled internal standards (100 ng/mL each). After solid-phase extraction, the analytes were eluted into 1 mL of methanol and then diluted to 100 μL with 5% methanol containing 1 mM AA. Moreover, 25 µL of the aliquot was injected during LC-MS/MS.

### 2.5. Analytical Method for Urinary Malondialdehyde

Thiobarbituric acid reacted with MDA to form a color compound, which could be detected colorimetrically at 530–540 nm. Calibration solutions of five concentrations were used, namely, 1.25, 2.5, 5, 10, and 25 μM, with the 5 μM samples used as the spike samples. The inter- and intra-assay coefficients of variation (and SD) were 5.1% (0.24) and 7.6% (0.45), respectively. Malondialdehyde levels in urine samples were measured using a commercial kit (Cayman Chemicals, No. 10009055, Ann Arbor, MI, USA).

### 2.6. Analytical Method for Urinary Phthalate Metabolites

The analytical methods for urinary phthalate metabolites have been validated in our previous studies [26,27,28,29]. The analyzed phthalate metabolites were as follows: MEHP, mono (2-ethyl-5-oxo-hexyl) phthalate (MEOHP), MEHHP, mono (2-ethyl-5-carboxypentyl) phthalate (MECPP), mono (2-carboxymethylhexyl) phthalate (MCMHP), mono-iso-nonyl phthalate (MiNP), MBzP, mono-iso-butyl phthalate (MiBP), mono-n-butyl phthalate (MnBP), MEP, and monomethyl phthalate (MMP). In summary, an ultrasonic-assisted solid-liquid extraction method was used to measure the phthalate levels in urine samples, accompanied by the addition of ammonium acetate (AA, 20 µL, >98%, Sigma Aldrich, St. Louis, MO, USA), β-glucuronidase (10 µL, *E. coli* K12, Roche Biomedical, Mannheim, Germany), and a mixture of 10 isotopic (^13^C_4_) phthalate metabolite standards (100 µL, Cambridge Isotope Laboratories, Andover, MA, USA). An online system coupled with a liquid chromatography/electrospray tandem mass spectrometer was used (LC-ESI-MS/MS) (Agilent 1200/API 4000, Applied Biosystems, Foster City, CA, USA). A detailed description of the analytical column gradient program can be obtained from our previous study [28]. In addition, a negative multiple-reaction monitoring model was used for mass spectroscopy.

The limit of detection (LOD) for 11 urinary phthalate metabolites ranged from 0.1 to 0.7 ng/mL. One blank, repeat, and quality control (QC) sample was included in each batch of analyzed samples. To maintain good performance of analysis, the concentration of blank samples should be below two times the detection limit. The spiked pooled urine samples were made using a mixture of phthalate metabolite standards (20–50 ng/mL) in each sample. Relative percentage differences in duplicate samples and recovery of the QC sample were required to be within ±30%. Concentrations less than LOD were analyzed as 1/2 LOD values [30].

### 2.7. Statistical Analysis

Demographic data are presented as the means and standard deviations for the continuous variables (age and BMI at enrollment) and as percentages for categorical variables (family history of asthma, tobacco exposure, carpet usage, pet raising, and annual family income). The detection rate was calculated as the percentage of the samples with detectable phthalate metabolites out of the total samples.

An index, ΣDEHPm, was used as the cumulative index of DEHP metabolites, which is the sum of the molar levels of MEHP, MEHHP, MEOHP, MECPP, and MCMHP, and another index, namely, ΣDBPm, was used, which is the sum of the molar levels of MiBP and MnBP. Both indexes are expressed in nmol/mL. Distributions of urinary phthalate metabolites, total concentrations of DEHP metabolites, and oxidative stress biomarkers were described using the detection rate, geometrical mean (GM), median, and interquartile range (IQR) for the case and control groups. Oxidative/nitrosative stress biomarkers were transformed using the natural logarithm (ln) to meet the normality assumption. Additionally, the Wilcoxon signed-rank test was used to calculate the difference in medians between the case and control groups. The association among high levels of oxidative/nitrosative stress biomarkers, high urinary phthalate metabolite levels (75th percentile), and asthma risk was analyzed using conditional logistic regression.

The selection of covariates was based on a literature review, as well as their availability and statistical significance in the models. Final models were adjusted for the covariates of urinary creatinine level, passive smoking during pregnancy, annual family income, and raising a furry or feathery pet.

## 3. Results

### 3.1. Study Participants

At the beginning of the recruiting process, we verified the completeness of the demographics of participants and the availability of urine samples. Next, of the 298 participants, 97—in whom only four phthalate metabolites were analyzed—were excluded. Of the remaining 201 participants in whom 11 phthalate metabolites were analyzed, 47 patients had doctor-diagnosed asthma (Figure 1). Each case was matched with three controls by the date of enrollment. The corresponding propensity scores met the criteria of SMD < 0.1 and VR: 0.5–2, indicating an appropriate balance of covariates. Finally, 41 children with asthma were matched with 111 children without asthma (controls).

The participants’ demographic characteristics are presented in Table 1. In addition to age and sex, no significant differences were observed in the BMI (case, mean: 16.8 kg/m^2^; control, mean: 16.5 kg/m^2^) of the study participants. A total of 152 children aged 3–18 years were included in this study. Children with asthma had a significantly higher prevalence of passive smoking exposure during pregnancy (60.0% vs. 40.0%, *p* = 0.039) and a lower annual family income than healthy controls (59.4% vs. 25.3%, *p* = 0.001).

### 3.2. Distribution of Urinary Phthalate Metabolites and Oxidative Stress Biomarkers

Comparisons of phthalate metabolites and oxidative stress biomarkers between patients with asthma and the healthy controls are presented in Table 2. The detection rates of phthalate metabolites in the case and control groups were similar and ranged from 59% to 100%, except for MiNP (<50%) and MBzP (~60%), which were considered as binary variables in further analysis. The detection rate of oxidative/nitrosative stress biomarkers was almost 100%, except for 8-NO_2_Gua, which also had detection rates of more than 87.8% and 95.5% in the case and control groups, respectively. The highest GM of urinary phthalate metabolite concentrations (ng/mL) were observed for MEHP (49.6), MnBP (45.2), and MECPP (42.5) in the control group and MECPP (57.4), MnBP (43.8), and MEHP (40.8) in the case group.

The median MBzP and MCMHP concentrations in the case group were significantly higher than those in the control group (3.94 vs. 2.52 ng/mL, *p* = 0.02 and 14.4 vs. 10.7 ng/mL, *p* = 0.088, respectively). However, ΣDBPm and ΣDEHPm were similar between the two groups (Table 2). No significant differences were observed between the two groups in most of the oxidative/nitrosative stress biomarkers, except for a slightly lower 8-OHdG level in the case group (median = 3.70 ng/mL) than in the control group (median = 4.14 ng/mL) (Table 3).

Conditional logistic regression analysis was used to assess the individual effects of phthalate metabolites and oxidative/nitrosative stress biomarkers on asthma, and the results are presented in Table 4. After adjustment for confounders, no significant association was noted between high phthalate metabolite levels and asthma risk. The odds ratio (OR) of high levels of phthalate metabolites in the control group showed a slightly higher trend than those in the case group, especially for MEHHP. Additionally, no significant association was noted between high levels of oxidative stress biomarkers and asthma risk (Table 5). Finally, we considered the influence of other covariates and evaluated the association between higher phthalate metabolite levels and higher oxidative/nitrosative stress biomarker levels, with the 75th percentile level of both as the cutoff (Table 6). Among the oxidative stress biomarkers, 8-OHdG showed a significantly positive association with several phthalate metabolites, such as MMP (adjusted OR (aOR): 3.40, 95% confidence interval (CI): 1.30–8.89)), MiBP (aOR: 3.49, 95% CI: 1.33–9.20), MnBP (aOR: 3.02, 95% CI: 1.08–8.47), MEHP (aOR: 2.66, 95% CI: 1.05–6.72), MEHHP (aOR: 2.82, 95% CI: 1.04–7.69), MECPP (aOR: 2.90, 95% CI: 1.08–7.80), MCMHP (aOR: 3.87, 95% CI: 1.40–10.7), MiNP (aOR: 2.79, 95% CI: 1.10–7.06), and ΣDEHPm (aOR: 3.44, 95% CI: 1.31–9.02). Among other oxidative stress biomarkers, 8-NO_2_Gua and 8-isoPF2α were associated with phthalate metabolites such as MMP (aOR: 2.66, 95% CI: 1.03–6.92 and aOR: 4.04, 95% CI: 1.51–10.8, respectively), and 8-isoPF2α was associated with MiBP (aOR: 2.96, 95% CI: 1.13–7.79) (Table 5 and Figure 2). To comprehensively analyze the joint effect of the 11 phthalate metabolites on asthma, we used a WQS regression model, which calculated both positive and negative associations weighted linear indices and identified the chemicals of concern in the mixture. The corresponding weight index shows the contribution of a particular phthalate to the WQS index [31].

Furthermore, WQS analyses were performed to examine the effects of phthalate metabolite mixtures on health outcomes. Figure 3 shows the results of the WQS regression analysis of the association between the combined 11 phthalate metabolites and asthma. As illustrated in Figure 2, the WQS index was not significant for asthma (0.88; 0.76–1.03, *p* = 0.115). MBzP contributed more than half of the association (56.8%), followed by MiBP (26.6%) and MiNP (8.77%) (Figure 3).

## 4. Discussion

Because asthma can be affected by certain demographics and environmental factors, we conducted a case-control study by using propensity score matching. For most phthalate metabolites, the median urinary phthalate levels in the case group were slightly higher than those in the control groups, especially for MBzP (*p* = 0.002). The increase in the association between higher phthalate metabolite levels also led to an increase higher oxidative stress and lipid peroxidation biomarker levels, such as 8-OHdG and 8-isoPF2α. However, no clear positive association was observed between the indicators of phthalate exposure and asthma. An obvious difference was noted in urinary phthalate levels between children with and without asthma, but the results were not significant. Alternatively, our data revealed more concerns regarding the relevant behavioral changes in participants who already have asthma or allergic conditions than in healthy people, and that exposure to well-known phthalate sources would be decreased in participants who already have asthma or allergic conditions. In addition, certain phthalates or their metabolites might function as adjuvants for different agents that cause allergic diseases.

Hoppin et al. reported an association between the sum of DEHP metabolites and allergic sensitization in adults [32]. Moreover, participants with both symptoms and allergic sensitization had higher urinary MBzP levels than those without sensitization or with symptoms alone. In another study, the association between phthalate exposure and asthma in adolescents was not observed using the oxidative stress biomarker 8-OHdG as a mediator [33]. However, patients with atopic keratoconjunctivitis exhibited higher lipid peroxidation, such as hexanoyl lysine (HEL) and 4-hydroxynonenal (HNE) [34]. Other in vitro studies have reported that lipid peroxidation could cause immune and airway inflammatory responses [35,36,37].

In an epidemiological study, Beko et al. divided participants into cases and controls and further evaluated the differences in phthalate exposure by allergic or nonallergic symptoms [38]. In the nonallergic asthma group, the daily DnBP exposure dose in patients with asthma was lower than that in controls (median: 0.74 vs. 0.98 µg/kg/day). The level of DnBP in house dust in patients with asthma was also lower than that in the controls (median: 7.7 vs. 14.9 µg/g), and the median urinary phthalate concentrations were also lower than in healthy controls (MnBP: 58.4 vs. 87.6 ng/mL; MEHP: 3.7 vs. 5.18 ng/mL; MECPP: 29.9 vs. 36.6 ng/mL). In the nonallergic group, the median DnBP concentrations in dust collected in the daycare center of rhinoconjunctivitis patients were lower than those in healthy individuals (DnBP: 14.2 vs. 32.0 µg/g; DiBP: 15.2 vs. 22.4 µg/g). Similar to our study, lack of allergic symptoms at the time would indicate lower exposure. In the allergic group, daily exposure to BBzP in participants with asthma was higher than that in the controls (median: 0.045 vs. 0.029 µg/kg/day), and the median phthalate dust concentration in the daycare center was higher in patients with asthma than in the controls (DnBP: 40.0 vs. 32.0 µg/g; BBzP: 22.1 vs. 15.2 µg/g). However, the median urinary MiBP level in patients with asthma was lower than that in the controls (44.3 vs. 75.6 ng/mL). Compared with the controls, patients with allergic rhinoconjunctivitis were exposed to higher phthalate dust concentrations in the daycare center (median: DnBP: 42.6 vs. 32.0 µg/g; DiBP: 25.4 vs. 22.4 µg/g). However, total BBzP exposure in the allergic group was higher in patients with atopic dermatitis (median: 0.047 µg/kg/day) than in the controls (median: 0.029 µg/kg/day). After adjustment for sex, breastfeeding, socioeconomic status, genetic susceptibility to allergy, and interior decoration, the asthma risk in the third tertile of DnBP was 7.9 times higher than that in the first tertile (OR: 7.9, 95% CI: 1.7–35.9).

Several studies have focused on education of parents and children regarding a particular environment intervention. In fact, more than 80% of parents of asthmatic children could identify at least one environmental asthma trigger, and 82% of these parents attempted to control such triggers in a survey recruiting a nationwide sample of 896 children (2–12 years) with asthma [39]. Otherwise, these interventions have been operating at a relatively low cost and acquired the best adherence, such as routine cleaning (e.g., mattress encasements, washing bedding, vacuuming) [40]. Several studies have also found that phthalates have been measured in dust samples from different microenvironments, including homes [41,42], daycare centers [43], schools [44], and workplaces [45]. Therefore, these suggested home-cleaning intervention strategies could effectively reduce potential dust and phthalate exposures.

These studies inferred that, compared with healthy people, people who already have allergic diseases will implement relevant behavioral changes to control the allergic conditions, leading to a decrease in exposure. Alternatively, certain phthalates or their metabolites might function as adjuvants for different agents that cause allergic diseases.

In the present study, we divided our participants into high- and low-concentration groups according to the median and 75th percentile, but the prevalence of high concentrations of phthalates in the asthma group was lower than that in the control group, which is in agreement with the findings of Beko et al., 2015 in the nonallergic group. Furthermore, Beko et al., 2015 observed lower concentrations of DEHP metabolites in the asthma group, which is consistent with our results [38]. However, we recruited children and adolescents aged from 3 to 18 years, whereas Beko et al., 2015 included only 3–5-year-old children, and the exposure pathway may not have been consistent. For example, they observed that breastfeeding was an important factor, and other studies have demonstrated that asthma in children under 7 years of age is more closely associated with maternal prenatal exposure [38].

Similar to our study results, Callesen et al. [8] discovered that decreasing asthma risk was observed in the fourth quartile of MnBP and MiBP relative to the first quartile. However, the relationship was not significant after adjustment for covariates. Ait Bamai et al. also reported that high exposure to DiBP led to a decreased risk of wheezing in boys (OR: 0.62, 95% CI: 0.40–0.97), and that increased exposure to DEHP was associated with a reduced risk of atopic dermatitis in boys (OR: 0.45, 95% CI: 0.21, 0.95), but not in girls [46]. Even if the study observed that dust containing phthalates caused asthma, it is more likely that the dust itself is a disease-inducing allergen [47]. Urinary phthalate metabolites are regarded as the sum of phthalate exposure. Therefore, these measurements could not provide any information corresponding to the contribution of different exposure pathways [48,49]. The abovementioned results support the hypothesis that inhalation and dermal absorption occurring in the indoor environment have more significant contributions to health than total exposure, which includes food ingestion.

Ketema et al. reported that only HEL, but not 4-hydroxynonenal (HNE) and 8-OHdG, was marginally positively associated with wheezing and eczema [50]. Ketema et al. reported that phthalate metabolite exposure was significantly associated with allergies, but no significant association was observed between phthalate metabolites and oxidative stress biomarkers. Moreover, lipid peroxidation in patients with asthma was positively associated with asthma severity in another study [51]. Several studies have reported an association between phthalates and oxidative stress, especially for 8-OHdG in children [52,53,54].

In the present study, we provide novel evidence with a well nested case-control study design with propensity score matching to understand the associations between urinary phthalate metabolites and risk of asthma attack, utilizing a series of measures commonly used to assess oxidative/nitrosative stress biomarker among children with asthma. These biomarkers of effect may assist in elucidating the underlying mechanisms of oxidative stress, a well-recognized factor in asthma responses, which may play a role in the adjuvant pathway of phthalate-induced asthma responses. However, there is no clear mediating effect of the oxidative stress biomarkers on the association between phthalate exposure and asthma.

Asthma is a chronic inflammatory airway disease, and oxidative stress is believed to play an important role in its pathogenesis; especially ROS have an essential role in the signaling pathways of antigen-induced allergic inflammation and many respiratory diseases including asthma [55]. Oxidative stress is one possible pathway for the DEHP-adjuvant effect in its contribution to the increased incidence of asthma and allergies. In animals, repeated exposure to DEHP had an adjuvant effect on the rat and mouse immune system response to being co-exposed to the allergen ovalbumin (OVA) in sensitized animals by gastric gavage at a dose of 70 mg/kg/day or 0.03 mg/kg/day [16,56]. You et al. have suggested that DEHP could enhance the sensitization of BALB/c mice to allergens and further upregulate the ROS level in pulmonary tissues. However, the DEHP exposure alone group does not show significant airway structural change and inflammatory cell infiltration, compared with the control group [57].

In the present study, asthma was diagnosed by a well-trained physician, obviating concerns about misclassification. Our study used various strategies to evaluate the matching criteria of the control group, thus retaining a larger sample size while ensuring successful matching.

This study has some limitations. First, the sample size was small. To decrease the contribution of selected covariates, we conducted a case-control study with propensity score matching. We observed some significant associations between oxidative stress biomarkers, including 8-NO_2_Gua and 8-isoPF2α, and phthalate metabolites, such as MMP, as well as between 8-isoPF2α and MiBP. Second, the case-control design precluded the determination of causality. Although the half-life of phthalates is short, a single morning urine sample may be representative [7,8,58]. Several studies have reported an association between allergic diseases such as allergic rhinitis and atopic dermatitis and phthalates [6,7]. However, the strength of the association between phthalate exposure and health outcomes did not change after including these allergic diseases in our study. Third, even we did not measure the phthalate levels in house dust collected in the child’s home directly, from previous qualitative studies [43,59], parents of children with asthma reported that they increased their use of air purifiers and dehumidifiers. After the implementation of Taiwan’s food safety incident, house dust containing phthalates is the major exposure factor at present; thus, parents’ behavioral changes may reduce children’s exposure to phthalates [60].

## 5. Conclusions

Consistent with the results of several studies, the median MBzP concentrations in the case groups were significantly higher than those in the control group. After adjustment for confounders, no significant association was noted between higher phthalate metabolite levels and asthma risk. However, some oxidative stress biomarkers, including 8-NO_2_Gua and 8-isoPF2α, were associated with phthalate metabolites such as MMP. An individual’s sensitization status may elucidate the somewhat conflicting evidence regarding the associations between phthalate exposure and asthma. Moreover, the direct associations between phthalate metabolites and asthma could not be observed in the present study, possibly because of the adjuvant effect of phthalates in enhancing the immune system response but not through the mediation of oxidative stress.

## Figures and Tables

**Figure 1 biomedicines-10-01438-f001:**
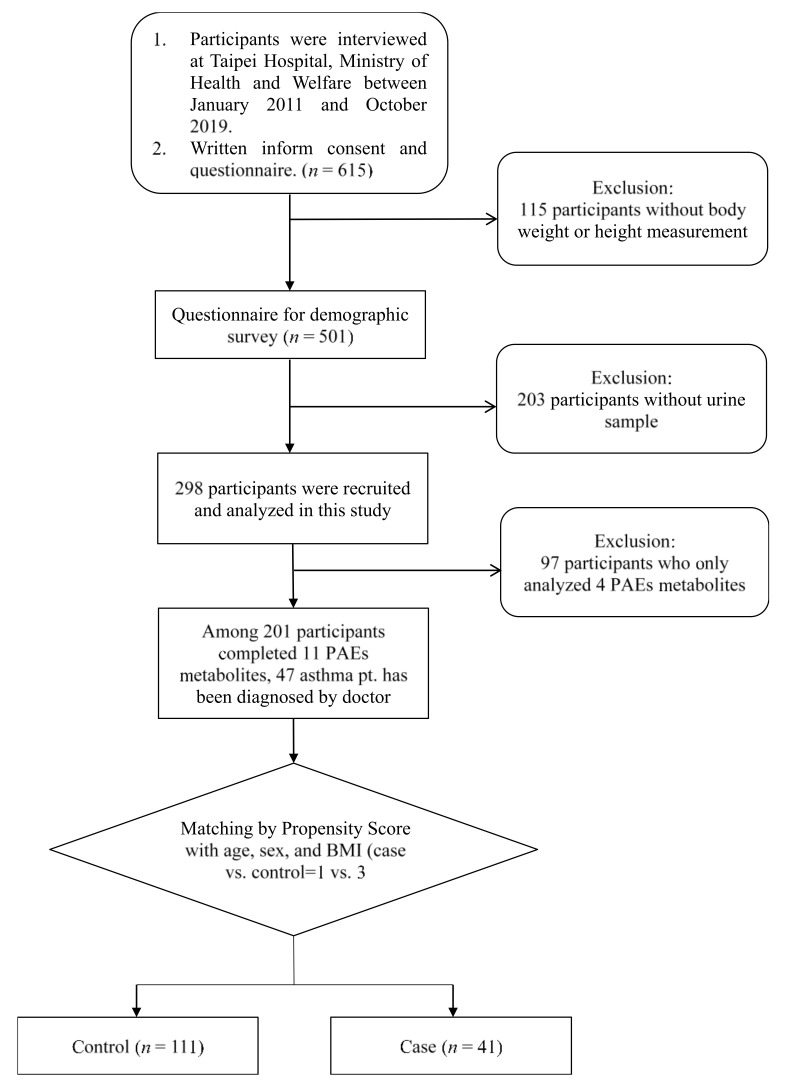
Flowchart of the recruitment of the study.

**Figure 2 biomedicines-10-01438-f002:**
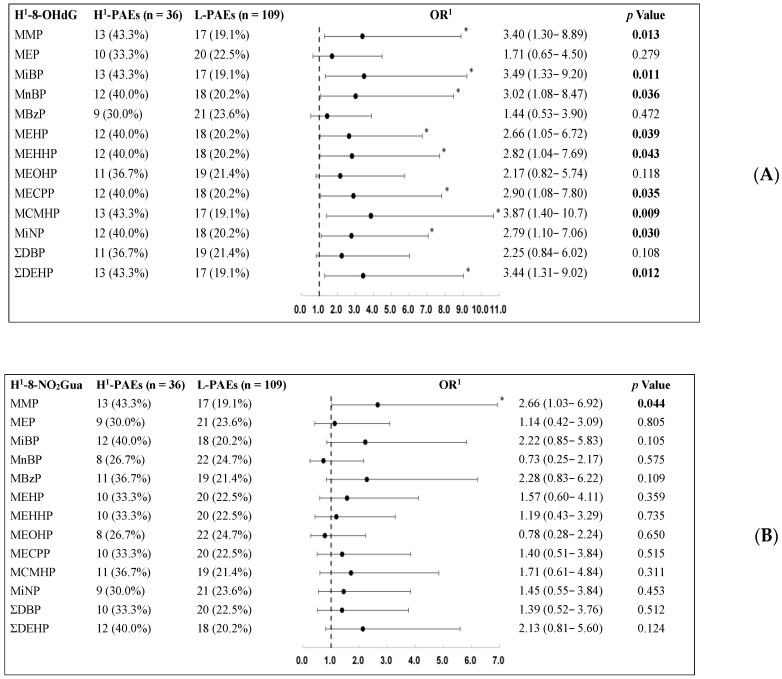
(**A**) Adjusted odds ratio for high oxidative/nitrosative stress in the different urinary phthalate metabolite level groups. H^1^: 75th percentile of the 8-OHdG and urinary phthalate metabolite level. Model adjusted for creatinine, passive smoking during pregnancy, annual income, and raising a furry or feathery pet. (**B**) Adjusted odds ratio for high oxidative/nitrosative stress in different urinary phthalate metabolite level groups. H^1^: 75th percentile of the 8-NO_2_Gua and urinary phthalate metabolite level. Model adjusted for creatinine, passive smoking during pregnancy, annual income, and raising a furry or feathery pet. (**C**) Adjusted odds ratio for high oxidative/nitrosative stress in different urinary phthalate metabolite level groups. H^1^: 75th percentile of the HNE-MA and urinary phthalate metabolite level. Model adjusted for creatinine, passive smoking during pregnancy, annual income, and raising a furry or feathery pet. (**D**) Adjusted odds ratio for high oxidative/nitrosative stress in different urinary phthalate metabolite level groups. H^1^: 75th percentile of the 8-IsoPF2α and urinary phthalate metabolite level. Model adjusted for creatinine, passive smoking during pregnancy, annual income, and raising a furry or feathery pet. (**E**) Adjusted odds ratio for high oxidative/nitrosative stress in different urinary phthalate metabolite levels groups. H^1^: 75th percentile of the MDA and urinary phthalate metabolite level. Model adjusted for creatinine, passive smoking during pregnancy, annual income, and raised a furry or feathery pet. Parameters showing statistical significance (*p* < 0.05) are highlighted in bold and with * asterisk.

**Figure 3 biomedicines-10-01438-f003:**
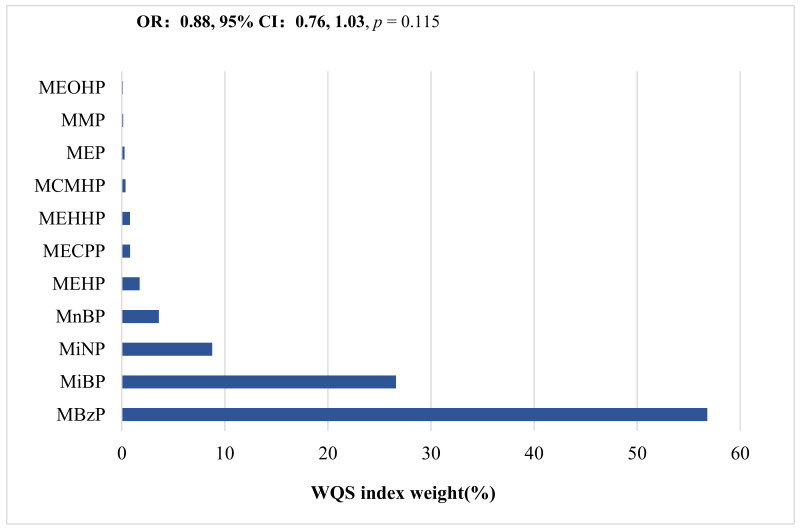
Weighted quantile sum (WQS) regression analysis of the association of urinary phthalate metabolites with asthma. Bar graphs show the magnitude of the WQS weight for each metabolite. Models are adjusted for creatinine, passive smoking during pregnancy, annual income, and raising a furry or feathery pet for the odds ratios (95% confidence interval) for association between phthalate metabolites and asthma.

**Table 1 biomedicines-10-01438-t001:** Comparison of the demographic characteristics between the case and control groups.

Demographic Characteristics	All (*n* = 152)	Control (*n* = 111)	Case (*n* = 41)	*p* Value ^c^
Sex				
Male	113 (74.3%)	82 (73.8%)	31 (75.6%)	>0.995
Female	39 (25.7%)	29 (26.1%)	10 (24.4)	
Age (years, Mean ± SD) (range)	7.4 ± 2.5 (3, 18)	7.3 ± 2.0 (3, 18)	7.5 ± 3.5 (3, 17)	0.759
BMI (kg/m^2^, Mean ± SD) (range)	16.6 ± 2.0 (13.2, 23.4)	16.5 ± 1.9 (13.2, 23.2)	16.8 ± 2.0 (13.4, 23.4)	0.245
Family history of asthma ^a^				
Yes	18 (12.3%)	11 (10.5%)	7 (17.1%)	0.276
No	128 (87.7%)	94 (89.5%)	34 (82.9%)	
Active smoking during Pregnancy ^a^				
Yes	4 (2.84%)	1 (0.98%)	3 (7.69%)	0.064
No	137 (97.2%)	101 (99.0%)	36 (92.3%)	
Passive smoke during Pregnancy ^a^				
Yes	64 (45.7%)	40 (40.0%)	24 (60.0%)	**0.039**
No	76 (54.3%)	64 (60.0%)	16 (40.0%)	
Child passive smoking exposure ^a^				
Yes	84 (59.6%)	58 (58.0%)	26 (63.4%)	0.577
No	67 (40.4%)	42 (42.0%)	15 (36.6%)	
Living closer to a major road ^a^				
Yes	90 (82.6%)	64 (83.1%)	26 (81.3%)	0.788
No	19 (17.4%)	13 (16.9%)	6 (18.8%)	
Lay carpet ^a^				
Yes	13 (9.2%)	10 (9.9%)	3 (7.5%)	0.759
No	128 (90.8%)	91 (90.1%)	37 (92.5%)	
Raise furry or feathery pet ^a^				
Yes	25 (17.5%)	17 (16.7%)	8 (19.5%)	0.808
No	118 (82.3%)	85 (83.3%)	33 (80.5%)	
Annual family income ^a,b^				
<600,000 NT	44 (33.9%)	24 (25.3%)	19 (59.4%)	**0.001**
≥600,000 NT	84 (66.1%)	71 (74.7%)	13 (40.6%)	

^a^ The sum is <100% because some participants did not answer this question.; ^b^ The currency exchange rate of converting USD to the new Taiwan dollar is 1:32; ^c^ Wilcoxon signed-rank test calculated for the difference in medians between the case and control groups. ^c^ Chi-squared test determines whether the distributions of categorical variables differ between the case and control groups. Parameters showing statistical significance (*p* < 0.05) are highlighted in bold.

**Table 2 biomedicines-10-01438-t002:** Distribution of urinary phthalate metabolites and oxidative/nitrosative stress biomarkers in the case and control groups.

Phthalate	Control (*n* = 111)				Case (*n* = 41)				
(ng/mL)	DR%	GM (GSD)	Range Min-Max	Median (IQR)	DR%	GM (GSD)	Range Min-Max	Median (IQR)	*p*
MMP	93	12.5 (5.81)	(ND, 1451)	14.1 (5.87, 34.9)	93	11.2 (4.91)	(ND, 160)	15.6 (4.61, 28.9)	0.874
MEP	88	15.3 (9.00)	(ND, 1676)	18.9 (7.21, 58.6)	93	14.0 (5.97)	(ND, 246)	18.8 (7.88, 42.9)	0.644
MiBP	97	25.9 (5.92)	(ND, 1919)	21.4 (8.72, 88.7)	100	28.9 (3.98)	(2.34, 1092)	20.4 (11.5, 96.0)	0.787
MnBP	98	45.2 (5.82)	(ND, 1555)	31.3 (15.4, 191)	100	43.8 (4.70)	(3.27, 812)	34.8 (15.7, 179)	0.936
MBzP	59	1.50 (9.10)	(ND, 1737)	2.52 (ND, 7.06)	61	2.16 (11.8)	(ND, 4548)	3.94 (ND, 11l3)	**0.020**
MEHP	94	49.6 (8.81)	(ND, 7855)	35.3 (20.0, 175)	95	40.8 (8.21)	(ND, 2282)	36.2 (16.3, 139)	0.535
MEHHP	95	29.3 (6.85)	(ND, 24521)	29.8 (9.84, 134)	100	35.2 (4.19)	(2.22, 344)	38.8 (11.0, 111)	0.608
MEOHP	86	12.7 (8.39)	(ND, 1540)	16.5 (5.81, 39.4)	87	13.7 (7.69)	(ND, 289)	21.9 (6.69, 61.2)	0.665
MECPP	95	42.5 (7.40)	(ND, 1599)	34.5 (13.0, 191)	100	57.4 (4.42)	(3.94, 775)	74.9 (13.9, 212)	0.577
MCMHP	91	9.96 (6.79)	(ND, 1354)	10.7 (4.50, 27.7)	90	9.75 (6.13)	(ND, 342)	14.4 (5.17, 27.0)	0.088
MiNP	50	2.69 (34.1)	(ND, 2294)	2.13 (ND, 67.6)	44	1.04 (14.8)	(ND, 300)	ND (ND, 15.7)	**0.004**
(nmol/mL)									
ΣDEHPm		0.81 (4.64)	(0.03, 84.5)	0.82(0.25, 2.62)		0.71 (4.42)	(0.04, 8.73)	0.79(0.20, 1.80)	0.874
ΣDBPm		0.36 (5.40)	(0.02, 8.73)	0.27(0.10, 1.14)		0.38 (4.25)	(0.03, 8.14)	0.26(0.14, 1.22)	0.949

ND: Data values below the LOD were replaced with LOD/√2; Wilcoxon signed-rank test calculated for the difference in medians between the case and control groups. Parameters showing statistical significance (*p* < 0.05) are highlighted in bold. Abbreviations: DR = detection rate; IQR = interquartile range; Min = minimum value; Max = maximum value; GM = geometric mean; GSD = geometric standard deviation; LOD = limit of detection, mono-methyl phthalate (MMP), mono-ethyl phthalate (MEP), mono-iso-butyl phthalate (MiBP), mono-n-butyl phthalate (MnBP), mono-benzyl phthalate (MBzP), mono-ethylhexyl phthalate (MEHP), mono-(2-ethyl-5-hydroxyhexyl) phthalate (MEHHP), mono-(2-ethyl-5-oxo-hexyl) phthalate (MEOHP), mono-(2-ethyl-5-carboxypentyl) phthalate (MECPP), mono-(2-carboxymethylhexyl) phthalate (MCMHP), mono-iso-nonyl phthalate (MiNP); ΣDBPm = MnBP + MiBP; ΣDEHPm = MEHP + MEOHP + MEHHP + MECPP + MCMHP.

**Table 3 biomedicines-10-01438-t003:** Distribution of urinary phthalate metabolites and oxidative/nitrosative stress biomarkers in the case and control groups (cont’d).

Biomarkers	Control (*n* = 111)				Case (*n* = 41)				
	DR%	GM (GSD)	Range Min-Max	Median (IQR)	DR%	GM (GSD)	Range Min-Max	Median (IQR)	*p*
MDA (μmol/L)	100	6.22 (1.94)	(2.00, 41.10)	5.86 (4.06, 8.30)	100	6.12 (1.93)	(1.15, 29.90)	6.36 (4.22, 8.50)	0.415
8-OHdG (ng/mL)	100	4.11 (1.87)	(0.92, 13.16)	4.14 (2.74, 6.38)	100	3.48 (1.90)	(0.94, 12.01)	3.70 (2.36, 5.16)	0.060
8-NO_2_Gua (ng/mL)	95.5	3.09 (2.02)	(ND, 8.74)	3.63 (1.95, 5.48)	87.8	2.64 (2.28)	(ND, 7.96)	3.29 (1.43, 4.93)	0.574
4-HNEMA	100	29.7 (1.81)	(6.18, 86.50)	29.0 (21.3, 45.3)	100	27.2 (1.91)	(5.89, 80.1)	28.6 (16.5, 40.6)	0.660
8-IsoPF2a	100	4.71 (1.76)	(1.00, 14.13)	4.85 (3.20, 7.03)	100	4.58 (1.99)	(1.30, 12.78)	4.78 (2.12, 8.12)	0.544

**Table 4 biomedicines-10-01438-t004:** Crude odds ratio (cOR) and adjusted odds ratio (aOR) for high urinary phthalate metabolite levels in the case and control groups.

	Case (*n* = 41)	Control (*n* = 111)	cOR 95%CI	*p* Value	aOR 95%CI	*p* Value
Above 75th percentile						
MMP	9 (22.0%)	29 (26.1%)	0.78 (0.35, 1.76)	0.551	0.39 (0.12, 1.32)	0.130
MEP	9 (22.0%)	29 (26.1%)	0.84 (0.37, 1.93)	0.679	0.49 (0.14, 1.67)	0.255
MiBP	11 (26.8%)	27 (24.3%)	1.14 (0.48, 2.69)	0.768	0.73 (0.22, 2.40)	0.599
MnBP	10 (24.4%)	28 (25.2%)	0.96 (0.43, 2.13)	0.920	0.30 (0.07, 1.32)	0.111
MEHP	9 (22.0%)	29 (26.1%)	0.82 (0.35, 1.97)	0.663	0.38 (0.10, 1.44)	0.153
MEHHP	8 (19.5%)	30 (27.0%)	0.63 (0.26, 1.52)	0.303	0.30 (0.74, 1.23)	0.096
MEOHP	13 (31.7%)	25 (22.5%)	1.57 (0.70, 3.55)	0.276	0.56 (0.15, 2.08)	0.389
MECPP	11 (26.8%)	27 (24.3%)	1.11 (0.49, 2.50)	0.807	0.50 (0.13, 1.94)	0.315
MCMHP	9 (22.0%)	29 (26.1%)	0.78 (0.32, 1.89)	0.583	0.29 (0.07, 1.28)	0.102
ΣDEHP	9 (22.0%)	29 (26.1%)	0.79 (0.34, 1.87)	0.593	0.11 (0.09, 1.27)	0.107
ΣDBP	11 (26.8%)	27 (24.3%)	1.14 (0.50, 2.60)	0.750	0.51 (0.11, 2.33)	0.383
Above LOD						
MBzP	25(61.0%)	65(58.6%)	1.11(0.53, 2.31)	0.781	0.85(0.32, 2.26)	0.740
MiNP	18(43.9%)	56(50.5%)	0.82(0.40, 1.68)	0.581	0.25(0.20, 1.53)	0.251

Analyzed with conditional logistic regression and using the control group as the reference group. aOR: adjusted for creatinine, passive smoking during pregnancy, annual income, and raising a furry or feathery pet. High urinary phthalate metabolite level = number above the 75th percentile or detection limit of urinary phthalate metabolite level.

**Table 5 biomedicines-10-01438-t005:** Crude odds ratio (cOR) and adjusted odds ratio (aOR) for high oxidative/nitrosative stress biomarkers in the case and control groups.

Oxidative/Nitrosative Stress Biomarkers	Case (*n* = 41)	Control (*n* = 104)	cOR 95%CI	*p* Value	aOR 95%CI	*p* Value
Above 75th percentile						
8-OHdG	8 (19.5%)	28 (26.9%)	0.67 (0.27, 1.67)	0.389	0.90 (0.29, 2.81)	0.853
8-NO_2_Gua	8 (19.5%)	28 (26.9%)	0.63 (0.24, 1.60)	0.328	1.07 (0.29, 4.01)	0.917
HNE-MA	10 (24.4%)	26 (25.0%)	1.00 (0.45, 2.25)	0.995	0.96 (0.34, 2.68)	0.930
8-IsoPF_2α_	11 (26.8%)	25 (24.0%)	1.20 (0.52, 2.75)	0.667	1.78 (0.61, 5.23)	0.291
MDA	11 (26.8%)	25 (24.0%)	1.15 (0.50, 2.63)	0.747	0.98 (0.28, 3.44)	0.978

Analyzed with conditional logistic regression and using the control group as the reference group. aOR: adjusted for creatinine, passive smoking during pregnancy, annual income, and raised a furry or feathery pet.

**Table 6 biomedicines-10-01438-t006:** Adjusted odds ratio for high oxidative/nitrosative stress biomarkers in the different urinary phthalate metabolite level (75th percentile) groups.

Odds Ratio	8-OHdG		8-NO_2_Gua		HNE-MA		8-IsoPF2α		MDA	
	aOR	95%CI	aOR	95%CI	aOR	95%CI	aOR	95%CI	aOR	95%CI
MMP	**3.40**	**(1.30, 8.89)**	**2.66**	**(1.03, 6.92)**	0.83	(0.29, 2.40)	**4.04**	**(1.51, 10.8)**	0.92	(0.32, 2.66)
MEP	1.71	(0.65, 4.50)	1.14	(0.42, 3.09)	0.60	(0.20, 1.76)	1.76	(0.67, 4.64)	0.93	(0.33, 2.64
MiBP	**3.49**	**(1.33, 9.20)**	2.22	(0.85, 5.83)	0.45	(0.14, 1.42)	**2.96**	**(1.13, 7.79)**	1.63	(0.60, 4.41)
MnBP	**3.02**	**(1.08, 8.47)**	0.73	(0.25, 2.17)	**0.22**	**(0.06, 0.85)**	0.94	(0.32, 2.74)	1.58	(0.56, 4.44)
MBzP	1.44	(0.53, 3.90)	2.28	(0.83, 6.22)	0.70	(0.24, 2.07)	2.12	(0.81, 5.58)	1.26	(0.44, 3.62)
MEHP	**2.66**	**(1.05, 6.72)**	1.57	(0.60, 4.11)	1.75	(0.68, 4.48)	1.16	(0.44, 3.07)	1.21	(0.43, 3.39)
MEHHP	**2.82**	**(1.04, 7.69)**	1.19	(0.43, 3.29)	0.57	(0.18, 1.78)	1.45	(0.51, 4.09)	0.72	(0.24, 2.20)
MEOHP	2.17	(0.82, 5.74)	0.78	(0.28, 2.24)	0.59	(0.19, 1.82)	1.80	(0.67, 4.82)	1.13	(0.40, 3.21)
MECPP	**2.90**	**(1.08, 7.80)**	1.40	(0.51, 3.84)	0.77	(0.26, 2.29)	2.17	(0.81, 5.84)	1.34	(0.47, 3.78)
MCMHP	**3.87**	**(1.40, 10.7)**	1.71	(0.61, 4.84)	1.03	(0.35, 3.03)	2.38	(0.86, 6.57)	0.72	(0.22, 2.32)
MiNP	**2.79**	**(1.10, 7.06)**	1.45	(0.55, 3.84)	1.46	(0.56, 3.80)	1.16	(0.44, 3.09)	1.14	(0.41, 3.16)
ΣDBP	2.25	(0.84, 6.02)	1.39	(0.52, 3.76)	0.30	(0.09, 1.07)	1.72	(0.64, 4.63)	1.81	(0.67, 4.89)
ΣDEHP	**3.44**	**(1.31, 9.02)**	2.13	(0.81, 5.60)	1.89	(0.71, 5.08)	1.94	(0.72, 5.19)	1.25	(0.44, 3.55)

Analyzed through multiple logistic regression and using the control group as the reference group. aOR: adjusted for creatinine, passive smoking during pregnancy, annual income, and raising a furry or feathery pet. Parameters showing statistical significance (*p* < 0.05) are highlighted in bold.

## Data Availability

All donors involved in this study signed an informed consent form before the sample collection.

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
