# Peer review of "Phthalate Exposure and Oxidative/Nitrosative Stress in Childhood Asthma: A Nested Case-Control Study with Propensity Score Matching"

_biomedicines, 2022, doi:10.3390/biomedicines10061438_

Round 1

Reviewer 1 Report

In the manuscript “Phthalate Exposure and Oxidative/Nitrosative Stress in Childhood Asthma: A Nested Case–Control Study with Propensity Score Matching”, the authors proposed to evaluate the individual and mixed effects of low-dose phthalate exposure on children with asthma and five mechanistic biomarkers using a propensity score–matched case-control study. The paper is somewhat descriptive without a clear rationale for the experimental approach, particularly for oxidative stress biomarkers.. Discussion should be presented in a more straightforward way. I have some specific comments that may be useful.

Specific comments:

1. The rationale for the experimental design is not clear nor the clinical relevance of this work. It seems a bit preliminary without a mechanistic approach. However, the authors could speculate a bit regarding the mechanisms mediating the observed effects.

2. The clinical significance of the data should be on spotlight. In addition, the oxidative stress biomarkers data should be well discussed. This could enrich the discussion.

3. The conclusion: “… the adjuvant effect of phthalates in enhancing the immune system response but not through the mediation of oxidative stress..” deserves first a good discussion.

Author Response

as attachment 

Reviewer 2 Report

The manuscript entitled “Phthalate Exposure and Oxidative/Nitrosative Stress in Childhood Asthma: A Nested Case–Control Study with Propensity Score Matching” by Jung-Wei Chang et al. examined the association of various urinary phthalate metabolites with asthma diagnosis. The materials and methods sections were clearly written and the analysis methods employed were appropriate. The overall findings of this study was negative. Still, this paper may be of interest to the journal’s targeted audience as its finding is consistent with other reports. However, the reviewer has some major concerns and a couple of minor concerns as outlined below.

Major concern:

(1) The second and third paragraphs of the Introduction should be rewritten. The first paragraph did a good job describing the prevalence of asthma in the general population in Taiwan and a need to investigate the underlying mechanisms, but the following paragraphs failed to lay down the rationale of selecting the five biomarkers for the present study. Some information was presented but not in a cohesive fashion. For example, what is the logic link between lines 70-74 and lines 75-77? The authors stated that 8-OHdG is a biomarker for DNA damage, but then what is 8-NO2Gua for? Another example is lines 77-80, the authors need to clarify the specific settings of the cited studies and also explain what kind of conclusions were drawn from these studies. Jus saying “.... have been evaluated simultaneously” was too vague, thus lacking relevance.

(2) In Discussion, the author stated that “The increase in the concentration of phthalates also led to an increase in oxidative stress injury indicators, such as 8-OHdG and 8-isoPF2α”. The reviewer could not find such data (Table 2), instead, the opposite. The authors need to clarify how this conclusion was drawn.

(3) Lines 325-330, what are the basis of these speculations? Did the authors have data on behavioral changes of participants? What is the literature support? Particularly, why “exposure to well-known phthalate sources would be decreased in participants who already have asthma or allergic conditions” lines 327-328? Some relevant information to answer the above questions were presented in “Strength and Limitations”, these information should be presented earlier in Discussion to support the authors’ statement, lines 325-330.

Minor concerns:

There are places that appear to be confusing to the reviewer, which is likely to be so to the readers. Some examples are listed below.

(1) The biomarker malondialdehyde first appeared in the Abstract, in full spelling without an abbreviation, but its second appearance in the Introduction was with an abbreviation (MDA) (page 2, line 65). The abbreviation should be with the first appearance.

(2) Page 2, line 64, what is DBP? It should be spelled out for its first appearance.

Author Response

as attachment 

Round 2

Reviewer 1 Report

The revised version of the manuscript has improved.